# Comparative Assessment of Different Pre-Treatment Bonding Strategies to Improve the Adhesion of Self-Adhesive Composites to Dentin

**DOI:** 10.3390/polym14193945

**Published:** 2022-09-21

**Authors:** Magali Inglês, Joana Vasconcelos e Cruz, Ana Mano Azul, Mário Polido, António H. S. Delgado

**Affiliations:** 1Instituto Universitário Egas Moniz (IUEM), Monte de Caparica, 2829-511 Almada, Portugal; 2Centro de Investigação Interdisciplinar Egas Moniz (CiiEM), Monte de Caparica, 2829-511 Almada, Portugal; 3Division of Biomaterials & Tissue Engineering, UCL Eastman Dental Institute, University College London, Royal Free Hospital, Rowland Hill Street, Hampstead, London NW3 2PF, UK

**Keywords:** biomaterials, dentin bonding, microtensile bond strength, polymerization, self-adhesive composites, light microscopy

## Abstract

The aim of this study is to compare the adhesive interface formed in dentin, using self-adhesive composites applied with different bonding strategies, by testing the microtensile bond strength (μTBS) and ultramorphology through the use of light microscopy. Permanent, sound human molars were randomly allocated to six experimental groups. The groups included a negative control group, where only etching was performed via EtchOnly; a positive control group where an adhesive was applied, OptiBondFL (OBFL); and an experimental group where a primer was applied using a co-curing strategy together with a composite (Primer_CoCuring). The samples were sectioned into microspecimens for μTBS (*n* = 8) and into 1-mm thick slabs for light microscopy using Masson’s trichrome staining protocol (*n* = 3). The statistical analysis included a two-way ANOVA for μTBS data and Tukey’s HSD was used as a post-hoc test (significance level of 5%; SPSS v. 26.0). The results of the μTBS revealed that the self-adhesive composite (F = 6.0, *p* < 0.018) and the bonding strategy (F = 444.1, *p* < 0.001) significantly affected the bond strength to dentin. However, their interactions were not significant (F = 1.2, *p* = 0.29). Etching dentin with no additional treatment revealed the lowest μTBS (VF_EtchOnly = 2.4 ± 0.8 MPa; CC_EtchOnly = 2.0 ± 0.4 MPa), which was significantly different from using a primer (VF_CoCu = 8.8 ± 0.8 MPa; CC_CoCu = 6.3 ± 1.0 MPa) or using the full adhesive (VF_OptiBondFL = 22.4 ± 0.3 MPa; CC_OptibondFL = 21.2 ± 0.4 MPa). Microscopy images revealed that the experimental Primer_CoCuring was the only group with no collagen fibers exposed to the dentin–composite interface. Overall, the use of a primer, within the limitations of this study, increased the bonding of the self-adhesive composite and provided sufficient infiltration of the collagen based on light-microscopic imaging.

## 1. Introduction

Resin composites are the preferred direct restorative material used in modern dentistry. These materials are esthetic polymer blends strengthened by inorganic glass fillers, resulting in a paste that is typically light-curable, on demand [1]. Composites exhibit fairly good mechanical behavior, with verified similarities to enamel and dentin [2], and excellent optical properties. However, the restorative procedure carried out with these materials still features several shortcomings [3]. Factors such as polymerization shrinkage, a higher susceptibility to biodegradation in the oral environment compared to inorganic materials, and clinicians’ fatigue are known problems associated with resin composites [4,5,6]. Furthermore, composites require a wearying and sensitive step-by-step process in order to achieve a stable bond to the tooth substrates and their correct application by incremental layering in the cavity.

Upon using a dental adhesive and depending on the adhesive strategy chosen, clinicians generally have to carry out application steps such as acid-etching, rinsing, drying, and the primer/adhesive’s application itself, which can come in two separate bottles [7]. All of this requires a skilled operator, a controlled environment to minimize the chances of contamination (i.e., with respect to blood and saliva), and the patient’s collaboration for an extended time. Moreover, by increasing the number of steps to comply with, there is a greater chance of introducing errors into each one of these steps, damaging the bonding procedure [8]. For these reasons, simplicity is often attained in adhesive dentistry. This search for easier and quicker material options has led to the development of self-adhesive composites, a relatively recent generation of composites that combines the capacity of a final restorative material with adhesive properties [9,10].

Self-adhesive composites have been developed and promoted as a hybrid all-in-one flowable composite with a chemical adhesion potential [11], and commercial options of these materials are currently available. The key difference of these materials compared to conventional composites is the functional monomer that is added to the composition, and its acidic moiety, which can vary. The main problem is that this class of composites remains unpopular due to its unsubstantiated claim of self-adhesiveness. Consistent in vitro and in vivo reports and pooled meta-analytical data have clearly shown that these composites do not establish a comparable bond to dental hard tissues, neither in the short- nor long-term, when compared to a conventional treatment with an adhesive and composite [12,13]. These latest composites do not have the ability to form a true hybrid layer in dentin and are therefore lacking in three-dimensional micromechanical interlocking features and collagen envelopment [14]. Self-evidently, their viscosity does not allow them to correctly wet and infiltrate the collagen network. The addition of a functional monomer is simply not enough to secure a durable bonding mechanism. Some authors have pointed out that the use of these materials requires careful individual consideration [15]. Thus, the improvement of current formulas is needed to achieve the next-generation of improved self-adhesive composites.

It is important to understand which surface treatments and modifications can secure the immediate bonding of these materials so as to optimize them; however, this has not been achieved yet. Earlier reports with self-adhesive cements, the precursors of self-adhesive composites, have shown that applying a primer coat before using these materials significantly improves their immediate bond strength to dentin [16,17]. In these cases, the primer achieves what the cement cannot: its solvated mixture expands the collagen network and acts as a carrier for hydrophilic monomers, ultimately increasing the surface free energy of dentin [18]. This, in turn, makes the surface receptable to a viscous resin. The self-adhesive composite would then function as a filled adhesive resin, placed on top of the primer [19]. 

Additionally, the co-curing technique has been proposed before in flowable composites [20,21]. This would grant the bypass of another step in the bonding procedure. It features the simultaneous curing of the primer and the flowable composite, which would allow the operator to skip a step, saving time and simplifying the procedure. However, it has shown conflicting results over the years [21,22,23]. In fact, neither the use of a primer by itself nor its co-curing with a self-adhesive composite have been investigated. This would provide an insight into optimizing the use of these materials. Thus, this study investigates the benefit of adding a primer step and its co-curing together with two different commercial self-adhesive composites with respect to the immediate bond strength to dentin and the microstructure of the interface that is formed.

## 2. Materials and Methods

### 2.1. Sample Acquisition and Materials

For this laboratory study, approved by the Ethics Committee of Instituto Universitário Egas Moniz (Protocol no. 1063), 66 permanent, sound human molars, which had been recently extracted, were gathered. These were free of carious lesions, restorations, or other structural defects, and were obtained with the patient’s consent. After being scaled and cleaned, the teeth were stored in 1% chloramine T (*v*/*v*) at 4 °C for one week, and then stored in distilled water, replaced every week, until they were used. All the molars were used within 6 months after their extraction.

The materials and their composition used throughout this study and the manufacturers’ details are shown in Table 1.

### 2.2. Experimental Design

For the bond strength study, 48 teeth were randomly assigned to six different groups (n = 8), depending on the composite used and the bonding strategy employed (Figure 1 and Table 2). As for the bonding strategy variable, a group in which only etching was employed as a negative control group, while a group featuring a full conventional etch-and-rinse protocol was used as a positive control for all composites tested (Table 2).

### 2.3. Sample Preparation and Restorative Procedure

The teeth were firstly sectioned parallel to the occlusal surface, beneath the cusps, using a low-speed diamond saw (Accutom-50, Struers A/S, Ballerup, Denmark), operating at 0.350 mm/s, and with water cooling applied to expose the mid-coronal dentin (Figure 2).

After sectioning, the smear layer was artificially simulated (600-grit SiC paper, Buehler Ltd., Lake Bluff, IL, USA) under running water for 60 s, using an automatic-polishing machine (LabolPol-4, Struers A/S, Ballerup, Denmark). According to the experimental group allocation, the samples were restored as described in Table 2.

Self-adhesive composite build-ups, using each assigned material, were carried out by incrementally adding 2 mm layers. Each layer was light-cured for 20 s, at minimal tip distance (~0 mm) using an LED light-curing unit (DB686, Froshan COXO Medical Instruments, Guangdong, China), operating at a mean irradiance at light exit of 900 mW/cm^2^, and with a wavelength range of 420–480 nm. The irradiance was measured with an analog radiometer (Optilux Radiometer SDS/Kerr, Orange, CA, USA) after every four consecutive uses. Each sample was left with a 6 mm self-adhesive composite build-up that was confirmed with a periodontal probe. The full restorative procedure is shown in Figure 3, Figure 4 and Figure 5.

### 2.4. Microtensile Bond Strength of Resin Composite–Dentin Interfaces

After storage for 24 h in deionized water in an incubator (37 °C at 100% humidity), the specimens used to test microtensile bond strength (μTBS) were sectioned longitudinally in the mesiodistal (X) and buccal–lingual (Y) directions, under running water, across the bonded interface, and using a hard tissue microtome (Accutom-50, Struers A/S, Ballerup, Denmark). This process resulted in rectangular composite–dentin-bonded beams—excluding the composite-enamel-bonded sticks—with a cross-sectional area of 1.0 ± 0.2 mm^2^, measured using a digital caliper (Storm digital caliper, CDC/N 0 150 mm, Pontoglio, Italy). 

The microtensile strength test was strictly performed in compliance with ADM guidance recommendations [24]. A universal testing machine (Shimadzu, Autograph AG-IS, Tokyo, Japan) with a load cell of 5 kN was used, operating at a crosshead speed of 0.5 mm/min. The μTBS (MPa) was calculated by dividing the load at failure by the cross-sectional bonding area (mm^2^). The failures were classified either as cohesive (failures that occurred exclusively within dentin or the resin composite), adhesive (failures that occurred exclusively at the composite/dentin interface) or mixed (failures that simultaneously occurred at composite/dentin interface and within dentin/composite). For statistical purposes, and following the ADM guidance, a value was attributed to pre-testing failures (PTF) as a mean between the lowest MPa value recorded for each group and 0 MPa.

### 2.5. Masson’s Trichrome–Light Microscopy

To further characterize the quality of the adhesive interface, 3 additional teeth were prepared for each self-adhesive composite experimental group, using the same procedure as the one described in Section 2.3 and Table 2. After 24 h storage (37 °C at 100% humidity), the samples from each group were again sectioned perpendicular to the bonded surface into 1 mm-thick slabs. In total, 9 sections were analyzed for each dentin treatment. Slabs were glued onto methacrylate supports with cyanoacrylate adhesive (Zapit, Dental Ventures of America, Corona, CA, USA) and ground down using a polisher machine (LabolPol-4, Struers A/S, Ballerup, Denmark) with SiC abrasive wet papers of descending grit size (320-, 600-, 1200-, and 4000-grit), until a thickness of 100 μm was achieved. Differential staining was accomplished with Masson’s trichrome, similarly to what was reported in Monticelli et al. (2008) [25]. The protocol used is shown in Table 3. After the coloration, samples were cover-slipped with mounting media and examined by light microscopy (CX41RF, Olympus, Tokyo, Japan) using 100× magnification. A photographic record was taken for each sample at low and high magnifications.

### 2.6. Statistical Analysis

In order to compare the microtensile bond strength means, inferential statistics tests were employed using the software Statistical Package for Social Sciences (SPSS) v. 26.0 for Mac (IBM Corporation, Armonk, NY, USA), at a set significance level of 5%. A two-way ANOVA model was used to factor in the two variables that were under study after performing normality and heteroscedasticity tests. Tukey’s HSD was used as a post-hoc test.

## 3. Results

### 3.1. Microtensile Bond Strength of Composite–Dentin Interfaces

The bond strength result means and the corresponding standard errors can be seen in the bar chart depicted in Figure 6, while the two-way ANOVA results are shown in Table 4. 

The two-way ANOVA results revealed that the self-adhesive composite used (F = 6.0, *p* < 0.018) and the bonding strategy employed (F = 444.1, *p* < 0.001) significantly affected the bond strength to dentin. However, their interaction was not significant (F = 1.2, *p* = 0.29). Etching the dentin only revealed the lowest bond strengths (VF_EtchOnly = 2.4 ± 0.8 MPa; CC_EtchOnly = 2.0 ± 0.4 MPa), which were overall significantly different to using a primer (VF_CoCu = 8.8 ± 0.8 MPa; CC_CoCu = 6.3 ± 1.0 MPa) and using the full adhesive (VF_OptiBondFL = 22.4 ± 0.3 MPa; CC_OptibondFL = 21.2 ± 0.4 MPa). Groups VF_EtchOnly were significantly different to VF_PrimerCoCu (Tukey’s HSD, *p* < 0.001), CC_PrimerCoCu (Tukey’s HSD, *p* = 0.003), VF_OBFL (Tukey’s HSD, *p* < 0.001), and CC_OBFL (Tukey’s HSD, *p* < 0.001). Similarly, CC_EtchOnly was also different to all other groups (*p* < 0.001), aside from VF_EtchOnly (*p* = 0.99). The groups where a primer was applied were not significantly different to each other (*p* = 0.13), nor were the groups in which the full etch-and-rinse protocol was undertaken with different self-adhesive composites, i.e., VF_OBFL vs. CC_OBFL (*p* = 0.76).

### 3.2. Bond Failure Mode Analysis

The failure mode and corresponding percentage, in relation to the global count of failures seen at the composite–dentin interface (adhesive, cohesive, and mixed), are depicted in Figure 7. In all the groups, the most predominant type of failure was the adhesive, as can be seen in the figure below.

The EtchOnly groups revealed the majority of pre-test failures (>50%), substantiating the low bond strength depicted in Figure 5. The groups where the primer and co-curing strategy were used, or where OptiBond FL was used, revealed far fewer pre-test failures and a higher adhesive failure distribution among the samples. 

### 3.3. Masson’s Trichrome–Light Microscopy

Light microscopy images resulting from Masson’s trichrome staining, which were divided and then subdivided between the different experimental groups, can be seen below in Figure 8.

When the conventional etch and rinse adhesive was used, a distinct red zone of denuded collagen could be seen at the base of the adhesive interface, although to a lesser degree than what was seen with the groups where only etching was carried out. Within the limitations of the technique, in the primer co-curing group, there was no evidence of incomplete collagen infiltration in the analyzed samples with its subsequent exposure, since the images were (Figure 7) all stained blueish green, without any red lines. 

## 4. Discussion

The natural evolution of resin-based materials—including adhesive systems, proven by many studies and being undertaken in this field, have allowed materials to become increasingly simplified, while still searching for ways to avoid compromising their effectiveness [20,26]. With the aim of minimizing chair time, technique sensitivity, and complex substrate handling, self-adhesive materials have emerged, and are now considered to be at the forefront of adhesive dentistry [18,27]. The adhesion of self-adhesive composites is specifically described as being based on the incorporation of acidic functional monomers. These may be monomers such as 10-MDP, GPDM, or others, such as 4-methacryloxyethyl trimellitic anhydride (4-META) or 2-methacryloxyethyl phenyl hydrogen phosphate (Phenyl-P), which have the ability to condition enamel and dentin, facilitating the creation of micromechanical retention zones and the eventual formation of chemical bonds between their functional groups and calcium from hydroxyapatite [18,28,29,30]. Although these strategies may respond to a clinical need, the interaction of these materials with dentin and their consequent bond strength is very limited, with insufficient interfacial interdiffusion [14]. Their poor retention and low sealing ability leads to high failure rates and the longevity of these self-adhesive composites is still very questionable [18,26,28,31].

Adhesion to enamel has proven to be strong and durable, since the simple, yet breakthrough discovery of the micromechanical interlocking in 1955 [18,32,33]. However, adhesion to dentin is much more intricate. In fact, it is only tangible when more complex, time-consuming, and technique-sensitive application procedures are followed [18,34]. Therefore, dentin was the focus of this study.

Currently, there are self-adhesive composites commercially available that reached the market a few years ago, such as VF and CC, which were used in this study. These self-adhesive composites have convergent practical indications. Due to their limited retention, these materials are only employed in the treatment of small, retentive, class I cavities as pit and fissure sealants, or as liners/bases of large class I and II restorations [28]. The hydrophobic–hydrophilic incompatibility between these resin-based materials and the dental substrate are obstacles in obtaining a strong and lasting bond strength, both short- and long-term [9,35]. This is one of the important issues experienced with these materials, although their unsuccess, especially with respect to the clinical failures witnessed, are due to multifactorial issues. This justifies the need to investigate and test new strategies that can improve their clinical behavior. The present study aimed to compare and evaluate the adhesive interface formed by using these two commercial flowable self-adhesive composites when different adhesive strategies are employed. 

Considering the results, the groups that reported the lowest microtensile bond strength values (<2.5 MPa), with the highest pre-test failure rates (71% and 51%), were the groups where solely etching was carried out. These poor results are due to the isolated application of orthophosphoric acid, known to affect the integrity of the dental organic matrix, whose structure corresponds to about 90% of type I collagen [36]. When the self-adhesive composite is placed over the conditioned substrate without a primer, due to its hydrophobic properties and overall viscosity, it will not be able to reach the full depth of demineralization, thereby compromising the restorative interface [18,33]. Poor infiltration and limited interaction lead to very low bond strength values, results that have also been corroborated in former studies that evaluated these materials [9,10].

To further simplify the restorative procedure, the co-curing technique was introduced. In this technique, the polymerization of the primer/adhesive and the self-adhesive composite occurs simultaneously [23,37,38]. Notably, these groups exhibited three-fold higher bond strength values than the groups where etching was carried out as a stand-alone treatment. Thus, the primer, due to its chemical composition, i.e., the presence of solvents and hydrophilic functional monomers capable of conditioning the dentin substrate and chemically binding to the calcium ions of hydroxyapatite (such as HEMA and GPDM in VF), provides advantages when the cavity involves dentin [18,29,33]. The light microscopy image results suggest an insufficient infiltration of the acid-etched demineralized dentin. Within the limitations of this investigation, this leads to the belief that the insufficient infiltration may be linked to viscosity and hydrophobicity–hydrophilicity incompatibility of the self-adhesive composite. The micromechanical retention guaranteed by the etching protocol alone, upon collagen exposure, did not allow for a correct hybrid layer formation, as hinted by Delgado et al. (2019) [31]. However, there are other materials and substrate characteristics that could impact the infiltration into the acid-etched demineralized dentin, and this warrants further research. These include the chemical composition of the composite, specifically, the type of monomers and their filler load and the moisture content of the acid-etched dentin [39,40]. Nevertheless, the primer alone and its co-curing potentially allowed for the formation of a hybrid layer, tripling the bond strength. The OBFL primer, despite being acidic due to the presence of the GPDM and PAMA monomers, is not a true self-etch primer, but rather an etch-and-rinse primer, which justifies the incomplete smear-layer removal and the low results compared to the full adhesive strategy [7,41,42,43]. In addition, the co-curing technique is not ideal for polymerization. The opacity and viscosity of the resin that comes on top—due to the presence of fillers—limits light penetration, which may not reach the depth of primer infiltration [44,45]. If the monomers are not fully polymerized, the cohesive properties of the adhesive interface are compromised, decreasing the bond strength, and affecting the overall adhesion mechanism [46].

The use of a primer as a pre-treatment strategy can be promising since it inevitably improves the wettability of hydrophobic resins. However, its chemistry must be optimized, and the polymerization technique must be improved to enable its application supported by the co-curing technique. Achieving this in the future may lead to higher bond strength values, while also limiting the hydrolytic degradation bound to occur in resin-based materials [36]. Different types of monomers used in the formulations of the primer, such as HEMA and GPDM, will influence the general properties of the resulting polymer, its bonding potential, and its susceptibility towards degradation. This affects the hybrid layer in dentin formed upon in situ polymerization [46,47]. The OptiBond^TM^ FL primer, with GPDM as its functional monomer, is able to infiltrate the water-rich collagen network. However, due to its hydrophilicity, also marked by HEMA, it is also prone to hydrolytic degradation, and forms semi-permeable membranes that can contribute to hybrid layers’ deterioration [48,49].

Although CC contains the functional monomer 10-MDP, as opposed to VF, which contains GPDM, both show similar results, contradicting the results obtained by Poitevin et al. (2013) [9]. Still, and as expected, the groups where the full adhesive system was applied showed the highest bond strength results. This is mainly because OBFL is claimed to be the gold standard of etch-and-rinse dental adhesives, with an outstanding and consistent performance with respect to bond strength and a hybrid layer quality verified regularly through in vitro and in vivo research [42,50,51]. The inter-diffusion zone of OBFL is known to be thick and uniform, as reported previously in the literature and corroborated in this study [52].

In agreement with Yuan et al. (2015), most of the failures are adhesive since this test method allows for the elimination of most of the cohesive failures in both dentin and resin [24,53,54].

To achieve self-adhesion, a relatively viscous resin composite must contain functional monomers capable of producing an effective ionic bond since they cannot penetrate deeply [9,53]. Thus, one of the strategies to improve the adhesion of these resins may involve increasing their interactive area by reducing their viscosity. This may be achieved by limiting the filler load of the mixture [44]. Yet, doing this will likely result in higher polymerization shrinkage, which is undesirable for bonding outcomes due to the generation of interfacial defects [55,56]. Thus, tailoring these material properties requires careful consideration to achieve a correct balance between the different properties if better adhesion is to be achieved. 

Masson’s trichome staining protocol enabled conclusions to be drawn regarding the identification of the microscopical layer constituents of the resin-dentin interface. This has been similarly accomplished in previous studies [57,58,59,60]. Based on the images obtained, it was possible to infer what was suspected in the microtensile bond strength results, discussed above. Indeed, the groups that used a separate etching step on its own revealed a higher thickness of the red-stained layer, indicating the presence of collagen fibrils that were not impregnated by resin and were thus exposed. This makes them susceptible to hydrolytic and enzymatic degradation, compromising the restorative interface as shown in the microtensile bond strength data over the years [47,61,62]. This layer was evidently thicker in these groups, proving that without a priming step, the self-adhesive composite cannot penetrate effectively and reach the demineralization depth. This may mean that the viscous material is collapsing and plasticizing the collagen further and forming an agglomeration of particles on top of the etch dentin, which makes it in turn even harder to infiltrate and envelop fibrils, justifying the low bond strength results [25,46,63]. In the OBFL group, a red stain is also visible. This is because the primer fails to reach the entire demineralization front, which can range from 8 μm to 10 μm, for 15 s of orthophosphoric acid (37.5%), again leaving some collagen fibrils bare [64].

In the groups where a primer was applied and co-cured, it is possible to observe dentin surfaces stained entirely in light green, which demonstrates, as expected, the absence of exposed collagen. The priming technique facilitates a simultaneous demineralization and penetration of the dentin surface, with a shallower hybrid layer formation [34,65]. By using an acidic primer to demineralize dentin and afterwards applying a self-adhesive flowable composite, we were able to polymerize both in situ, potentially guaranteeing an increase in marginal seal and reduction of hydrolysis mechanisms [29,65]. It is important to consider that in the two-bottle system, the adhesive part is supposed to act as a hydrophobic, impervious coating to protect the hybrid layer by preventing hydrolytic degradation [18,47]. In this case, the layer of self-adhesive composite, placed on top of the primer, potentially works even better as a bond-protecting layer and sealant since it is more hydrophobic.

Dentin is made up of a significant organic portion, and water, which is what makes bonding challenging [66]. During the bonding procedure, the same amount of water remaining in demineralized dentin, after etching, should be replaced by an equal amount of resin co-monomers. This is ideal, and this exchange, although not complete, is only possible because of the primer [67]. The primer increases the surface energy and receptivity of dentin to a hydrophobic resin, making hybridization possible [9,68,69]. This hybridization process does not occur with self-adhesive composites. Solvents play a vital role in granting the ability of the monomers to correctly interdiffuse in the water-rich organic content of dentin. They also act as diluents to reduce the overall viscosity of the resin. However, self-evidently, self-adhesive composites do not have solvents, which limits monomer diffusion into demineralized dentin. It is thought that the viscosity of these self-adhesive materials may be responsible for compressing and collapsing the collagen network, making it even more impenetrable. This explains why these flowable self-adhesive composites cannot, in any form, hybridize demineralized dentin on their own [70].

It is important to stress that the results obtained with the microscopical images are not free of limitations. They provide a qualitative assessment, and they should only be considered together with an appropriate quantitative analysis, as in this study, used to complement the bond strength data [25,57,58]. In the method used, interfacial defects, voids, or gaps smaller than 1 μm are undetectable. Thus, other imaging techniques such as TEM and SEM should be used to provide more information, with a higher resolving power; however, a sufficient bond is often needed to withstand sample preparation for these techniques, which is difficult to achieve with self-adhesive composites. For this reason, light microscopy was employed as an easier option.

Although the results show a tendency for improvement, when compared to the negative control where only etching was carried out, the self-adhesive composites still performed better when associated with an etch-and-rinse adhesive. Hence, it is important to investigate and combine the advantages of an acidic primer capable of demineralization and a self-adhesive composite that is fluid enough to enable an interaction between the resin monomers and the dentin substrate. This study is the starting point for further research and improvements of this material class. 

## 5. Conclusions

It was possible to conclude that the commercial self-adhesive composites benefit from application steps to achieve bonding to dentin. The best bond strength results were seen when these materials were used with an etch-and-rinse adhesive. However, placing these composites on a surface treated with a primer, in comparison to an etched surface only, was revealed to be more beneficial. This may suggest that, within limitations, factors such as the viscosity, the hydrophobicity–hydrophilicity mismatch, and the insufficient wettability of the substrate seen with these materials may be crucial problems. In fact, the co-curing strategy used showed less collagen exposure at the resin–dentin interface than other experimental groups. Thus, etching dentin to improve bonding does not seem to be indicated with these materials. It is important to continue to make efforts to improve this class of materials in order to enable quicker, more effective, and less technique-sensitive restorative treatments.

## Figures and Tables

**Figure 1 polymers-14-03945-f001:**
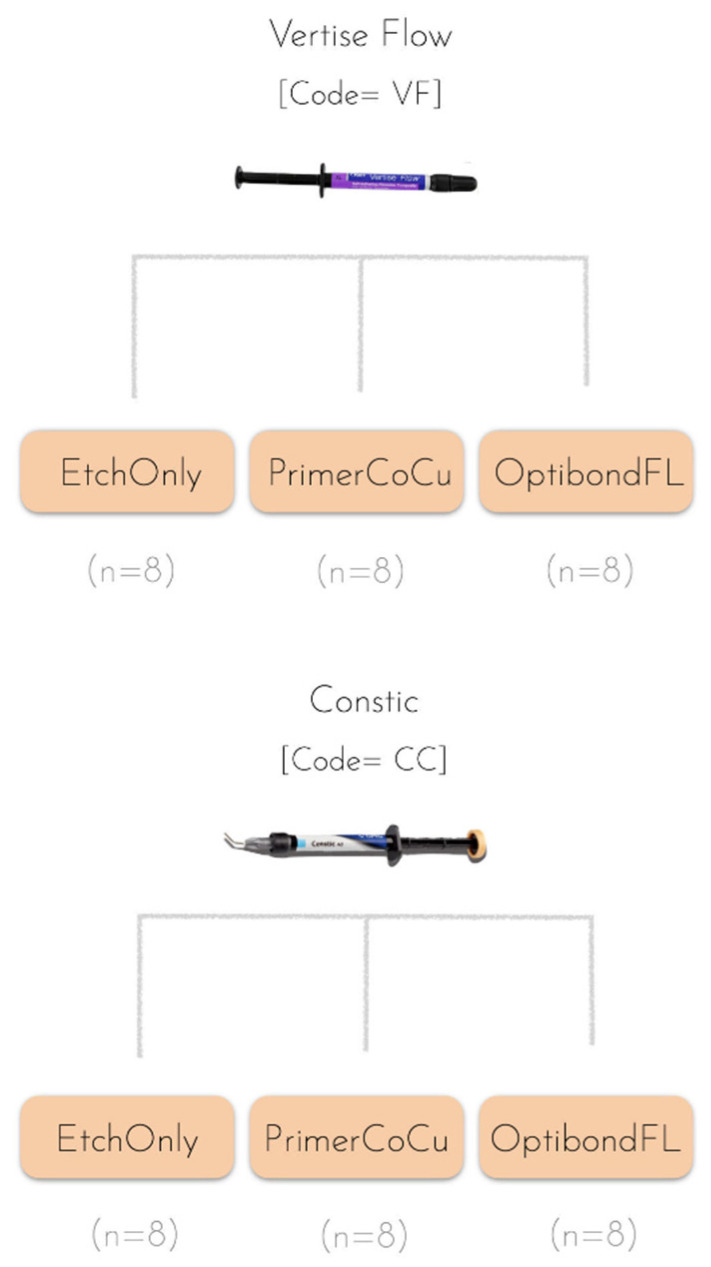
Experimental design followed in this study, with two distinct material groups, Vertise Flow (VF) and Constic (CC), each subdivided into three experimental groups based on the bonding strategy used (Table 2).

**Figure 2 polymers-14-03945-f002:**
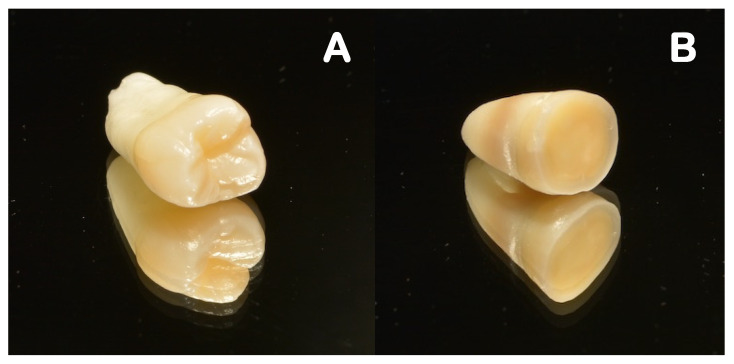
Example of a tooth selected for the study: (**A**) permanent and non-carious human molar; (**B**) longitudinally sectioned tooth.

**Figure 3 polymers-14-03945-f003:**
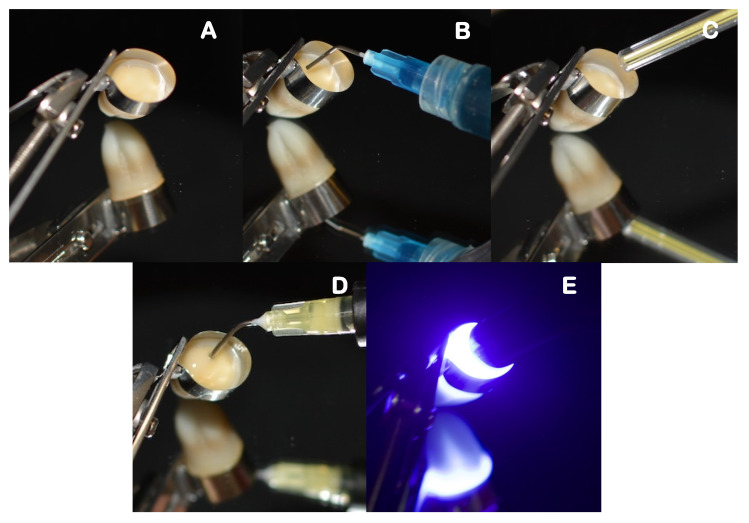
Restorative procedure of the etch-only group, where: (**A**) is the application of restorative matrix band, (**B**) etching protocol, (**C**) rinsing and drying step, (**D**) application of the flowable self-adhesive composite, and (**E**) light curing of each increment.

**Figure 4 polymers-14-03945-f004:**
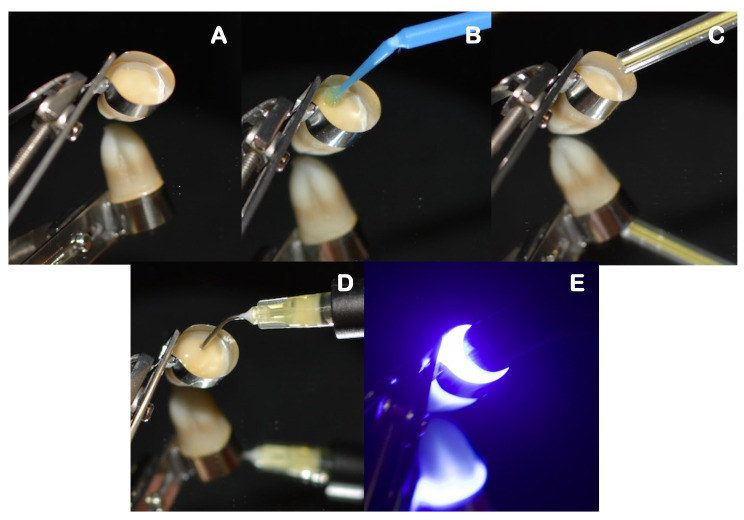
Restorative procedure of the primer co-curing group, where: (**A**) is the application of the restorative matrix band, (**B**) application of the primer, (**C**) drying step, (**D**) application of the flowable self-adhesive composite, and (**E**) light curing of each increment.

**Figure 5 polymers-14-03945-f005:**
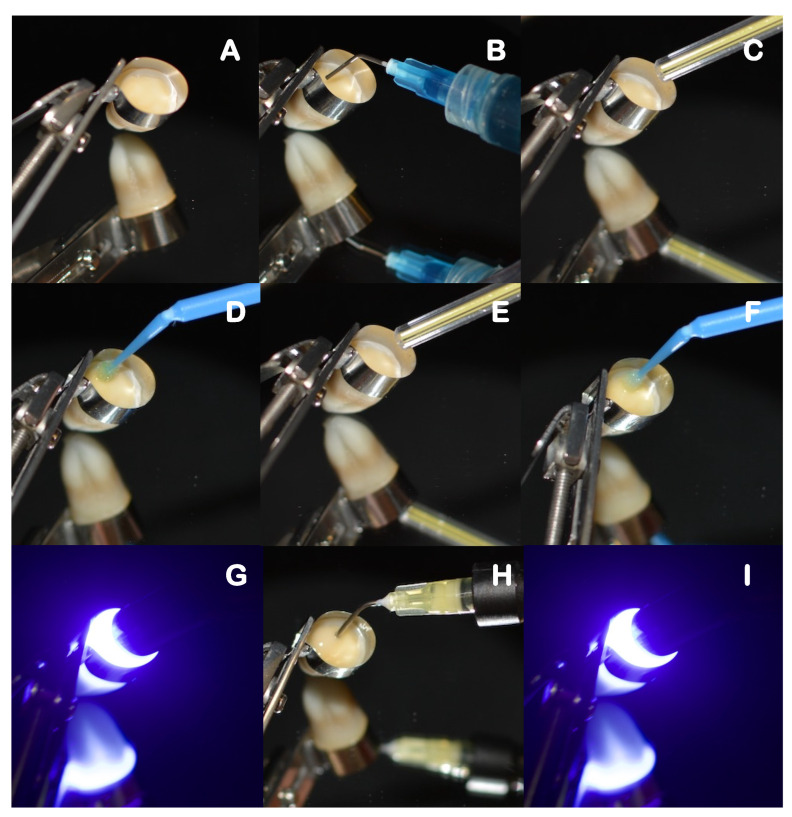
Restorative procedure of the Optibond FL (full etch-and-rinse) group, where: (**A**) is the application of the restorative matrix band, (**B**) etching protocol, (**C**) rinsing and drying step, (**D**) application of the primer (part I), (**E**) drying step, (**F**) application of the bond (part II), (**G**) light curing of the adhesive, (**H**) application of the flowable self-adhesive composite, and (**I**) light curing of each increment.

**Figure 6 polymers-14-03945-f006:**
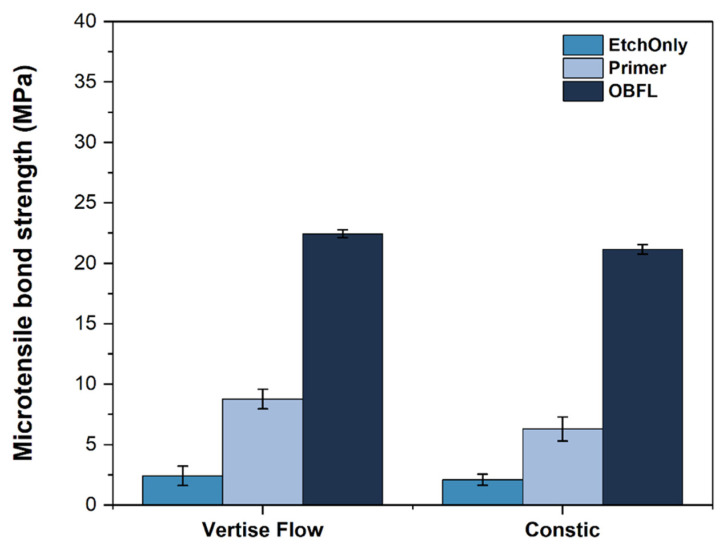
Bar chart comparing different groups in study. Etching dentin, as a stand-alone treatment, revealed the lowest bond strengths, which were significantly different to using a primer by itself and co-curing it together with the self-adhesive composite (Tukey’s HSD, *p* < 0.05). However, both of the groups were still significantly inferior to using the full etch-and-rinse protocol. Error bars shown are standard errors of means.

**Figure 7 polymers-14-03945-f007:**
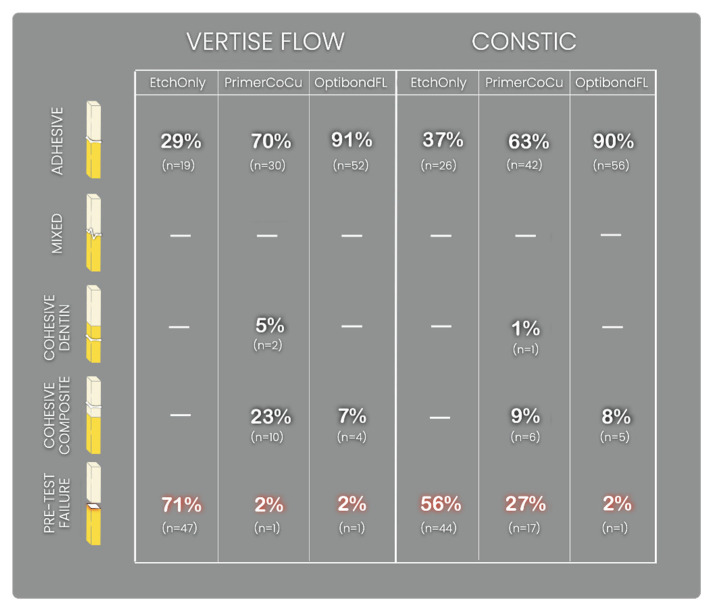
Summary of the fractographic analysis showing the different types of failures seen in each experimental group (presented as %). Pre-test failure indicates beams that fractured before bond strength tests were carried out.

**Figure 8 polymers-14-03945-f008:**
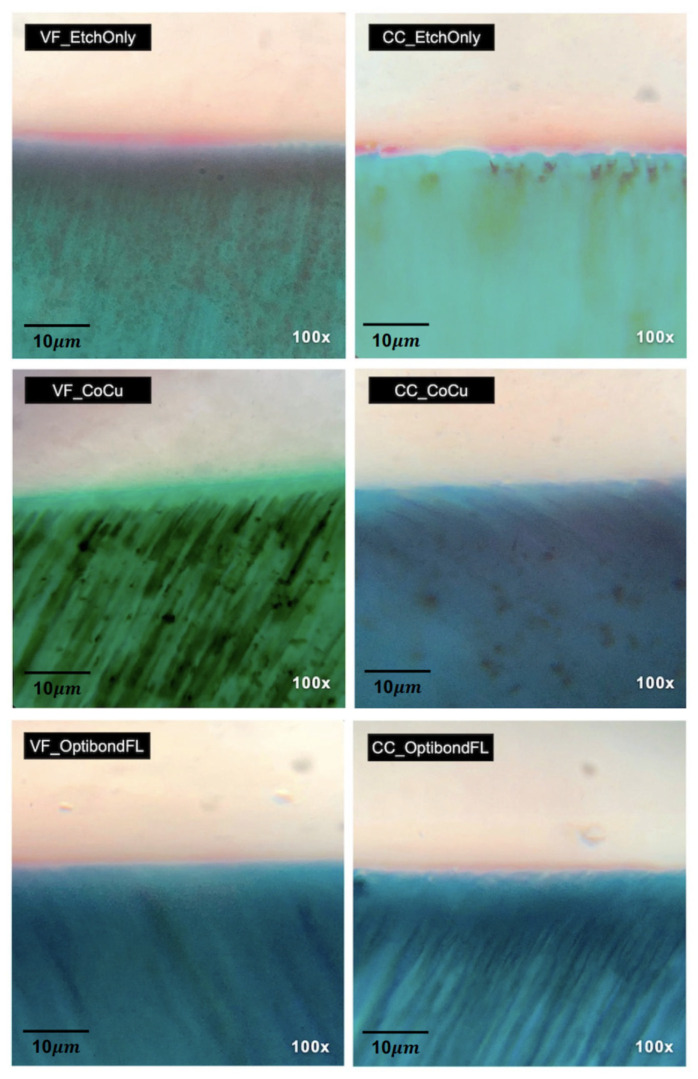
Representative light micrographs of the composite–dentin interfaces, stained with Masson’s trichrome, where: mineralized dentin can be seen in green; the self-adhesive composite or composite/adhesive is shown in light beige, with filler; while the exposed collagen is in red. A greater degree of exposed collagen can be clearly identified in sections belonging to specimens etched with orthophosphoric acid (VF_EtchOnly, CC_EtchOnly). Sections restored with OptiBond FL show less collagen exposure, but which is still visible, while primer and co-curing groups show no visible collagen exposure in the analyzed samples. Original magnification at 100× is shown.

**Table 1 polymers-14-03945-t001:** List of self-adhesive composites and dental adhesives used and their chemical compositions, according to information retrieved from the manufacturers and safety datasheets.

Material	Type	Manufacturer and Details	Composition
Vertise^TM^ Flow	Self-adhesive composite	Kerr Italia, S.r.I.; Scafati, Italy Shade: A2 Batch number: 8755817	**Organic matrix:** 5–10% HEMA, N/A% Bis-GMA, 5–10% UDMA, and 1–5% GPDM **Filler particles:** Ytterbium fluoride and barium aluminosilicate **Filler load:** 70 vol%
Constic	Self-adhesive composite	DMG Chem. -Pharm. Fabrik GmbH; Hamburg, Germany Shade: A2 Batch number: 256266	**Organic matrix:** 15–35% Bis-GMA, <45% TEGDMA, and N/A% 10-MDP **Filler particles:** Barium aluminosilicate **Filler load:** 66 vol%
OptiBond^TM^ FL Primer	Commercial etch-and-rinse dental adhesive (part I)	Kerr Italia, S.r.I.; Scafati, Italy Batch number: 7887502	10–30% HEMA, 5–10% GPDM, BHT, PAMA, CQ **Solvent:** Ethanol/Water
OptiBond^TM^ FL Bond	Commercial etch-and-rinse dental adhesive (part II)	Kerr Italia, S. r. I.; Scafati, Italy Batch number: 7517622	**Organic matrix:** 10–30% HEMA, N/A% Bis-GMA, GDMA **Filler particles:** Barium aluminosilicate, sodium hexafluorosilicate and fumed silica

10-MDP—10-methacryloyloxydecyl dihydrogen phosphate; BHT—2,6-di-(tert-butyl)-4-methylphenol; Bis-GMA—Bisphenol-A glycidyl dimethacrylate; CQ—1,7,7-trimethylbicyclo-[2,2,1]-hepta-2,3-dione; GPDM—glycerol phosphate dimethacrylate; HEMA—2-hydroxyethyl methacrylate; PAMA—Phthailic acid monomethacrylate; TEGDMA—triethylene glycol dimethacrylate; UDMA—Urethane dimethacrylate.

**Table 2 polymers-14-03945-t002:** Step-by-step procedure carried out during sample preparation, considering the three bonding strategies evaluated in this study.

Bonding Strategies		
Etch-Only	Primer Co-Curing	Optibond FL
Etching of the tooth surface for 15 s with 37.5% phosphoric acid Rinsing for 15 s with deionized water Drying for 15 s without dehydrating the surface of dentin Placement of the composite directly on the surface, with a thin layer (~1 mm) being applied first (VF and CC groups featured a rubbing motion on the dentinal surface for 15–20 s, according to manufacturer’s instructions). Light curing of each increment of 2 mm for 20 s, until a height of 6 mm was achieved.	Application of the Optibond FL primer (part I) on the surface, gently, for 20 s. Without rinsing, gentle drying was carried out using a mild air flow for 5 s. Placement of the composite directly on the surface, with a thin layer (~1 mm) being applied first (VF and CC groups featured a rubbing motion on the dentinal surface for 15–20 s, according to manufacturer’s instructions). Simultaneously light curing the primer with composite on top for 20 s. Light curing each increment of 2 mm for 20 s, until a height of 6 mm was achieved.	Etching of the tooth surface for 15 s with 37.5% phosphoric acid Rinsing for 15 s with deionized water Drying for 15 s without dehydrating the surface of dentin Application of the primer using a rubbing motion for 15 s. Gentle drying for 5 s. Application of the adhesive in a uniform, thin layer for 15 s. Light curing for 20 s. Placement of the composite directly on the surface, with a thin layer being applied first (VF and CC groups featured a rubbing motion on the dentinal surface for 15–20 s, according to manufacturer’s instructions). Light curing each increment of 2 mm for 20 s, until a height of 6 mm was achieved.

**Table 3 polymers-14-03945-t003:** Step-by-step procedure of Masson’s trichrome-staining protocol. Thorough washing was carried out before the protocol, in between dehydration solutions and in subsequent steps also to remove visible surface debris.

Reagents	Protocol
Solute A Weigert’s hematoxylin A + B in equal parts Solute B 2cc ponceau of 1% xylidine in 1% acetic acid or 1cc of 1% acid fuchsin in 1% acetic acid Solute C 1% phosphomolybdic acid Solute D 2% Light Green in 1% acetic acid	Dehydration using ethanol at 70%, 90%, and 100% for 20 min each Washing with distilled water for 15 s Solute A application for 30 min Washing with distilled water for 15 s Warm water for 3 min Immersion in solute B for 8 min Application of acetic acid for 10 s Washing with distilled water for 15 s Immersion in solute C for 40 min Application of acetic acid for 10 s Washing with distilled water for 15 s Contrasting with solute D for 2 min Application of acetic acid for 30 s Washing with distilled water for 60 s Dehydration with 70% Ethanol for 2 min Dehydration with 90% Ethanol for 2 min Washing with distilled water for 15 s

**Table 4 polymers-14-03945-t004:** ANOVA two-way output table, showing significance for both variables. Df represents degrees of freedom and Z.

	Type III Sum of Squares	df	Mean Square	Z	Sig.
Model	3302.921	5	660.584	179.386	<0.001
Intercept	5315.125	1	5315.125	1443.359	<0.001
Composite	22.277	1	22.277	6.049	0.018
Bonding Strategy	3271.365	2	1635.683	444.181	<0.001
Composite * Bonding Strategy	9.279	2	4.639	1.260	0.294
Pattern	154.664	42	3.682		
Total	8772.710	48			

## Data Availability

Data are available upon request from the corresponding author.

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
