# Peer review of "Comparative Assessment of Different Pre-Treatment Bonding Strategies to Improve the Adhesion of Self-Adhesive Composites to Dentin"

_polymers, 2022, doi:10.3390/polym14193945_

Round 1

Reviewer 1 Report

1. There are limitations in the sample preparation that must be addressed in the manuscript. As an example, polishing the specimens to achieve 100-micron-thick sections. The polishing can lead to the deposition of debris particles at the interface. Polishing can lead to artifacts that can interfere with accurate interpretation of the results. The authors must acknowledge this potential limitation and describe how they accounted for this limitation.

2. The authors indicate that the failures from the microtensile bond strength testing were classified as cohesive, adhesive or mixed. The authors must specify the technique, e.g., stereomicroscope, scanning electron microscope, and so forth, that was used to identify the bond failures.

3. Scale bars must be included on the light microscope images.

4. In the results, the authors report the following: “With the Primer CoCuring strategy there no sign of insufficient infiltration of collagen, with its subsequent exposure, in dentin, since images (Figure 7) all stained blueish green.”  This lack of evidence of collagen exposure with this treatment may be traced, in part, to the limitations of the technique. For example, the polishing protocol could lead to the deposition of debris that could be interpreted as representative of complete infiltration. In addition, the analytical technique inhibits the authors’ ability to detect voids that are below the resolution of the light microscope which is about 1 micron. These are very important limitations that must be discussed by the authors.

5. The authors suggest decreasing the viscosity of the self-adhesive composites by reducing filler loading. Reducing filler loading will likely lead to an increase in polymerization shrinkage. Polymerization shrinkage of the composite has been cited in numerous articles as a factor contributing to interfacial defects. The interfacial defects provide crevices that can be penetrated by bacteria, oral fluids and bacterial byproducts; penetration of pathogens at the interface between the composite and tooth will lead to secondary decay and failure of the composite restoration. The authors must discuss the potential detrimental impact associated with decreased filler loading of the composite.

6. The authors state that the “Masson’s trichome staining protocol allowed conclusions to be drawn regarding the ultramorphology and composition of the resin-dentin interface”. This is not an accurate statement. The authors used light microscopy to visualize the interface. The resolution of light microscopes is generally about 1 micron. The term ultramorphology is used to describe images that are acquired with transmission electron microscopy (TEM). TEM can image microstructural details at about 1nm to potentially a 0.1nm. The analytical techniques did not allow the authors to draw conclusions about ultramorphology. Furthermore, the staining revealed the presence of collagen, dentin, adhesive; the staining technique did not allow the authors to draw conclusions about the detailed composition of the interface.

7. The authors did not discuss the limitations of the investigation. As noted above, there are several limitations linked to the sample preparation and analytical techniques. These limitations must be addressed in the manuscript. Without appropriate discussion of these limitations, the conclusions could be misleading.  

Author Response

Point 1. There are limitations in the sample preparation that must be addressed in the manuscript. As an example, polishing the specimens to achieve 100-micron-thick sections. The polishing can lead to the deposition of debris particles at the interface. Polishing can lead to artifacts that can interfere with accurate interpretation of the results. The authors must acknowledge this potential limitation and describe how they accounted for this limitation.

Response: The reviewer is correct in pointing out this limitation, and we acknowledge this comment. We point out that we thoroughly washed each specimen, not only during the fixation procedure but also before we carried out the dehydration protocol and during. Many application steps included thorough washing that is known to remove the surface debris accumulated from polishing. This protocol was accomplished in many different studies and was stringently followed. This information was also added to the Table 3 caption.

Point 2. The authors indicate that the failures from the microtensile bond strength testing were classified as cohesive, adhesive or mixed. The authors must specify the technique, e.g., stereomicroscope, scanning electron microscope, and so forth, that was used to identify the bond failures.

Response: We acknowledge this lapse pointed out by the reviewer, which we have now corrected. We have added that the bond failures were classified using a stereomicroscope, in the microtensile bond strength methods section.

Point 3. Scale bars must be included on the light microscope images.

Response: We thank the reviewer for the suggestion - we have now added scale bars to the light microscopy images.

Point 4. In the results, the authors report the following: “With the Primer CoCuring strategy there no sign of insufficient infiltration of collagen, with its subsequent exposure, in dentin, since images (Figure 7) all stained blueish green.”This lack of evidence of collagen exposure with this treatment may be traced, in part, to the limitations of the technique. For example, the polishing protocol could lead to the deposition of debris that could be interpreted as representative of complete infiltration. In addition, the analytical technique inhibits the authors’ ability to detect voids that are below the resolution of the light microscope which is about 1 micron. These are very important limitations that must be discussed by the authors.

Response: We thank the reviewer for the suggestion. In combination with what was requested in point 7, we have now added a new paragraph to the discussion with the limitations of our study, to improve the discussion and better support the findings of the study.

Point 5. The authors suggest decreasing the viscosity of the self-adhesive composites by reducing filler loading. Reducing filler loading will likely lead to an increase in polymerization shrinkage. Polymerization shrinkage of the composite has been cited in numerous articles as a factor contributing to interfacial defects. The interfacial defects provide crevices that can be penetrated by bacteria, oral fluids and bacterial byproducts; penetration of pathogens at the interface between the composite and tooth will lead to secondary decay and failure of the composite restoration. The authors must discuss the potential detrimental impact associated with decreased filler loading of the composite.

Response: We thank the reviewer for the pertinent comment. We have now added a new section to the 9th paragraph of the discussion, to explain the problem of shrinkage and how these properties need to be tailored and balanced to achieve a decent material.

Point 6. The authors state that the “Masson’s trichome staining protocol allowed conclusions to be drawn regarding the ultramorphology and composition of the resin-dentin interface”. This is not an accurate statement. The authors used light microscopy to visualize the interface. The resolution of light microscopes is generally about 1 micron. The term ultramorphology is used to describe images that are acquired with transmission electron microscopy (TEM). TEM can image microstructural details at about 1nm to potentially a 0.1nm. The analytical techniques did not allow the authors to draw conclusions about ultramorphology. Furthermore, the staining revealed the presence of collagen, dentin, adhesive; the staining technique did not allow the authors to draw conclusions about the detailed composition of the interface.

Response: We acknowledge this mistake pointed out by the reviewer, which we overlooked initially. We have now changed the term "ultramorphology" and rephrased the sentence to "Masson's trichome staining protocol allowed conclusions to be drawn regarding the identification of microscopical layer constituents of the resin-dentin interface". We are aware that the detailed composition of the interface would only be possible with other methods such as vibrational spectroscopy methods (i.e. micro-Raman).

Point 7. The authors did not discuss the limitations of the investigation. As noted above, there are several limitations linked to the sample preparation and analytical techniques. These limitations must be addressed in the manuscript. Without appropriate discussion of these limitations, the conclusions could be misleading.

Response: We thank the reviewer for the suggestion. Accordingly, we have now added a new paragraph to the discussion with the limitations of our study, in order to support the present findings.

Reviewer 2 Report

Dear authors, it is well written and the topic is clinically relevant.

Please consider adding some bonding values to the abstract section and throughout the manuscript, please unite the used terminology, for example, either Resin composite Or Conventional composite Or Composite etc.

The introduction part is a bit long, please consider shortening and omitting the general paragraphs. Figure 6 needs some adjustment. 

I would rather use the term debonding failure analysis or failure mode analysis rather than fractographic analysis as no SEM  was used.

Shorten the conclusion please.

Author Response

We thank the author for the pertinent comments and suggestions and we are happy to know that the feedback is positive. We have made substantial changes to the manuscript to improve its scientific quality, and the points addressed in relation to what the reviewer requested can be found below.

Point 1. Please consider adding some bonding values to the abstract section and throughout the manuscript, please unite the used terminology, for example, either Resin composite Or Conventional composite Or Composite etc.

Response: We thank the reviewer for this, we have now added the MPa values to the abstract and manuscript, and we have united the terminology. However we still have places where we mention "resin composites" to generalize all composites and "self-adhesive composites" when we are mentioning the ones that are under study. 

Point 2. The introduction part is a bit long, please consider shortening and omitting the general paragraphs. Figure 6 needs some adjustment. 

Response: We thank the reviewer for the suggestion. We have cut some information out of the introduction and we have substantially rephrased many sentences in the introduction, while also trying to comply to other reviewer requests. We have also fully changed Figure 6 (failure mode analysis) as requested, and the number of the figure was wrong, so it is now Figure 7.

Point 3. I would rather use the term debonding failure analysis or failure mode analysis rather than fractographic analysis as no SEM  was used.

Response: The reviewer is right in pointing out this mistake. It has now been corrected to "bond failure mode analysis" in the results section.

Point 4. Shorten the conclusion please.

Response: We have revised the conclusion, also to meet the demands of other reviewers. We have cut some sentences and rephrased others to make it more conservative and supportive of the findings.

Reviewer 3 Report

This study aims to compare the adhesive interface of dentin prepared with different bonding strategies and distinct self-adhesive composites, by testing microtensile bond strength (mTBS) and the ultra-morphology using light microscopy. This study is well written, however, it needs to be improved, as follow:

L.12-14: “Groups included a negative control, […] a positive control group”. Please, specify these groups; it seems that were performed five groups as described in the present form.

L.34-35: “However, the restorative procedure carried out with these materials still features several shortcomings”. General information on the subject does not contribute to relevant knowledge. Therefore, develop the content addressing the limitations of this material.

L.38-49: “Conventional composites require multiple adhesive steps beforehand, even when using simplified adhesive versions, to ensure proper retention to enamel and dentin …”. The introduction of a new restorative material in the dental market is developed to overcome the limitations of previous materials, however, the devaluation of one material to value the next is not considered appropriate. The justification mentioned in relation to the multiple adhesive steps is in opposition to the objective of the study, which is the inclusion of one more step in the restorative technique, through a primer.

The methodological design of the present study presents a bias, to which no control group was added taking into account the manufacturer’s indications, that is, without the application of adhesive strategies prior to the application of the self-adhesive composite resin. Consider adding a positive control group following the manufacturer’s recommendations and its correct clinical use and indication.

Table 2: “Etched the surface for 15 s”. Be specific on which type of substrate the adhesive treatment is being carried out.

“Placed the composite directly on the surface, firstly a thin layer”. How many layers were inserted? How thick is this ‘thin layers’?

“Light cured each increment of 2 mm for 20 s.” This is the 9th step of Optibond FL.

Statistical analysis: Were normality and homoscedasticity tests performed?

Figure 5: Please, insert the representation of statistical symbols in the figure.

Figure 6: In addition to the percentages, include the absolute values (n=?) of each fracture type.

Figure 7: please, put the images in the same sequence as your study design.

L.384-385: “using a primer in co-curing mode seemed to be a promising strategy”. How is a promising strategy possible if the μTBS results were lower and less collagen exposure? Consider rewriting the conclusion with more appropriate definitions, based on your results.

References: review the standardization indicated by Polymers.

Author Response

We would like to thank the reviewer for the pertinent comments and suggestion that sparked changes in the manuscript which improved the overall scientific content of the present study. Below the reviewer can find their points addressed:

Point 1. L.12-14: “Groups included a negative control, […] a positive control group”. Please, specify these groups; it seems that were performed five groups as described in the present form.

Response: This has now been changed since it was, in fact, confusing. Changes can be seen in the same lines in the abstract.

Point 2. L.34-35: “However, the restorative procedure carried out with these materials still features several shortcomings”. General information on the subject does not contribute to relevant knowledge. Therefore, develop the content addressing the limitations of this material.

Response: We thank the reviewer for this suggestion. This content has now been developed to address the main limitations of composites, although it was kept short to meet the demands of other reviewers, that were concerned about the length of the introduction.

Point 3. L.38-49: “Conventional composites require multiple adhesive steps beforehand, even when using simplified adhesive versions, to ensure proper retention to enamel and dentin …”. The introduction of a new restorative material in the dental market is developed to overcome the limitations of previous materials, however, the devaluation of one material to value the next is not considered appropriate. The justification mentioned in relation to the multiple adhesive steps is in opposition to the objective of the study, which is the inclusion of one more step in the restorative technique, through a primer.

Response: We thank the reviewer for the comment, but we would like to clarify what we meant by the information provided in the text. Composites are excellent materials and the current adhesive restorative protocol is a sound protocol that works very well. Simplification of the protocol to achieve simpler restorations, that take less time, is however desirable by the patient and clinician. The objective of this study is in fact to see if a primer improves the bonding of self-adhesive composites, but not to advocate the use of a primer when these materials are used. It is merely from a research standview, which is very important to determine factors that help bonding of these materials. In the future this should be investigated thoroughly, in order to find the best pre-treatment strategy or solution that can aid the wettability of hydrophobic self-adhesive composites. Furthermore, even though we did add "one more step", we combined the light-curing of both the primer and the composite together, bypassing another step. So still, even when using this technique, it is much simpler than using a full adhesive strategy and normal composite layering with independent light-curing.

Point 4. The methodological design of the present study presents a bias, to which no control group was added taking into account the manufacturer’s indications, that is, without the application of adhesive strategies prior to the application of the self-adhesive composite resin. Consider adding a positive control group following the manufacturer’s recommendations and its correct clinical use and indication.

Response: We acknowledge the suggestion by the reviewer and we would like to point out that we have carried out tests using control groups that follow manufacturer recommendations, however, the bond strengths are so low or register 0 MPa as result of total pre-test failures that we are often unable to process data or include them in the statistical analysis. These results can also be commonly found in the literature. For the study design in this particular research, we felt it was more important to distinguish between a group where a full adhesive was used (Optibond FL), and between etching only, or applying a primer and co-curing only. This conveys the information needed to understand which is more important, as a surface pre-treatment, to secure immediate bonding of hydrophobic flowable composites.

Point 5. Table 2: “Etched the surface for 15 s”. Be specific on which type of substrate the adhesive treatment is being carried out. “Placed the composite directly on the surface, firstly a thin layer”. How many layers were inserted? How thick is this ‘thin layers’?. “Light cured each increment of 2 mm for 20 s.” This is the 9th step of Optibond FL.

Response: This has now been changed to "the tooth surface", since the etchant was added to the whole surface, but only dentin bonded sticks were tested. The layers, size of layers and the 9th step of Optibond FL were also added to Table 2.

Point 6. Statistical analysis: Were normality and homoscedasticity tests performed?

Response: Yes they were as they are mandatory in inferential statistics. These were added to the methods.

Point 7. Figure 6: In addition to the percentages, include the absolute values (n=?) of each fracture type.

Response: We thank the reviewer for the suggestion, figure 6 has now been changed and the absolute values of each fracture type were also added.

Point 8. Figure 7: please, put the images in the same sequence as your study design.

Response: Again, we thank the reviewer for the comments. We have changed the sequence of the images as suggested.

Point 9. L.384-385: “using a primer in co-curing mode seemed to be a promising strategy”. How is a promising strategy possible if the μTBS results were lower and less collagen exposure? Consider rewriting the conclusion with more appropriate definitions, based on your results.

Response: The conclusions have now been rewritten and reformulated as per suggestions. It seems a promising strategy since it achieved a bond strength three-fold higher than etching by itself. Also, we should consider less collagen exposure as a very positive sign. With less collagen exposure in dentin we will theoretically have less degradation since most fibrils will be encapsulated and protected by a resin-rich layer. It is a sign that the resin was able to correctly envelop the collagen network and that the demineralization achieved was shallow, which is desirable in dentin. This research opens the path for new primers or primer-like solutions that may be developed to aid bonding of self-adhesive composites.

Round 2

Reviewer 1 Report

Review of Revised Manuscript:

1. The last sentence of the abstract must be revised. The last sentence states: “Overall, the use of a primer increases bonding of the self-adhesive composites by achieving correct collagen envelopment in dentin.”

Based on the light microscopy images, it appears that the primer infiltrated the collagen and this infiltration inhibited Masson’s trichrome staining. However, it is not appropriate to conclude that this provided “correct collagen envelopment”. The current in vitro investigation studied the interface under static conditions. In the mouth, the interface will be exposed to a variety of chemical, physical and mechanical stresses. Masson’s trichrome stain does not provide a comprehensive representation of the molecules within the mouth. The molecular size of various components of oral fluids may be smaller than the Masson’s trichrome.

The statement that the treatment in this in vitro investigation led to “correct collagen envelopment” is misleading.

The last sentence must be revised to indicate that the primer increased bonding of the self-adhesive composite and provided sufficient infiltration of the collagen based on light microscopic imaging.

2. The last paragraph of the results section discusses the limitations. There is a clear disconnect between the results and information provided in the abstract. This disconnect must be corrected by revising the abstract to indicate the limitations of the investigation.

The last sentence of the results section indicates the following: “Within the limitations of the technique, in the Primer CoCuring group there was no evidence of incomplete collagen infiltration in the analyzed samples”.

3. The legend for Table 3 must be revised. In its current form, the legend indicates that the protocol “guaranteed effective surface debris removal”. The surfaces were not analyzed using various surface sensitive techniques, e.g., x-ray photoelectron spectroscopy, atomic force microscopy, and so forth.

It is misleading to report that the thorough washing guaranteed effective surface debris removal.

The legend must be revised to indicate that the surface was washed thoroughly to remove visible surface debris.

4. The authors must be commended for including positive and negative control groups. The control and experimental groups are clearly defined. The authors must also be commended for including pre-test failures (Figure 7). The large percentage of pre-test failures is very interesting. These pre-test failures highlight the challenges and limitations of bond strength testing.

5. In the discussion, the authors indicate that the hydrophobic-hydrophilic incompatibility is the central issue in the clinical failure of these materials. That is not an accurate statement. The clinical failure of these materials is a multi-factorial problem. Clinical failures have been linked to patient characteristics, technique sensitivity, the interplay of chemical and mechanical stresses at the interface, heterogeneity of the dentin substrate, and so forth. To describe hydrophobic-hydrophilic incompatibility as the central problem oversimplifies this complex, multi-factorial problem.  

The sentence must be revised to indicate that the hydrophobic-hydrophilic incompatibility is one of the problems leading to clinical failure of these materials.

6. The following statement (discussion section, rows 319-322) overstates the conclusions that can be drawn from the current investigation: “These results confirm that the main problem related to self-adhesive composites is their viscosity and hydrophobicity, since the micromechanical retention guaranteed by the etching protocol alone, upon collagen exposure, did not allow a correct hybrid layer formation.”

In the results section, the authors have identified a number of limitations in the current investigation. The results of this investigation do not support that the main problem of self-adhesive composites is viscosity and hydrophobicity. The authors did not measure these properties.

The sentence in the discussion is misleading and must be revised.

The light microscopy images suggest insufficient infiltration of the acid-etched demineralized dentin. Within the limitations of this investigation, the insufficient infiltration may be linked to viscosity and hydrophobicity of the self-adhesive composite. There are other material and substrate characteristics that could impact infiltration of the acid-etched demineralized dentin. There are recent review articles that discuss these characteristics. The following are offered as examples:  Jafarnia Sh, Zeinaddini Meymand J, Zandkarimi F, Saberi S, Shahabi S, Valanezhad A, et al. Comparative Evaluation of Microtensile Bond Strength of Three Adhesive Systems. Front Dent. 2022;19:8.

Kumar D, Bolskar RD, Mutreja I and Jones RS (2022) Methacrylate Polymers With “Flipped External” Ester Groups: A Review. Front. Dent. Med. 3:923780.

Stewart CA and Finer Y. Biostable, antidegradative and antimicrobial restorative systems based on host-biomaterials and microbial interactions. Dental Materials 35: 36-52, 2019.

A. Zhang, N. Ye, W. Aregawi, L. Zhang, M. Salah, B. VanHeel, H.P. Chew, and A.S.L. Fok. A Review of Mechano-Biochemical Models for Testing Composite Restorations. Journal of Dental Research 2021, Vol. 100(10) 1030–1038.

The results of this investigation cannot confirm that the main problem with self-adhesive composites is their viscosity and hydrophobicity. The investigators are drawing conclusions that cannot be supported by the results presented in this investigation.

7. The results of this investigation did not prove that the viscosity and insufficient wettability are the problem with the self-etch composites. The sentence in the conclusion section indicating that the study proved that these properties are the problem must be revised.

The sentence in its current form is not supported by the results presented in the current investigation.

Author Response

The authors would like to thank and acknowledge the comments and suggestions of the reviewer, for the second time, which again improved the scientific quality and soundness of the present paper. Please kindly find the replies below:

  1. We thank the reviewer for their comment, to which we agree. We have now changed the last sentence of the abstract to "increased bonding of the self-adhesive composite and provided sufficient infiltration of the collagen based on light microscopic imaging"
  2. Based on what was corrected in point 1, the abstract has now been revised to support what was written in the results. Many thanks for pointing this out.
  3. We thank the reviewer for revising the language and suggesting a new caption for Table 3, which we have now included. We have added the text according to the suggestion. Many thanks.
  4. Many thanks for this comment, we appreciate this fact.
  5. This sentence has now been revised to match the suggestion of the reviewer, to which we agree. It is indeed a multifactorial problem and the text has now been written and revised as such.
  6. We thank the reviewer for the suggestion and comment, once again. We understand that we were not being conservative and were proposing theories that cannot be confirmed based upon what was done in this study. We have now revised this sentence completely. We have changed it to: "The light microscopy image results suggest insufficient infiltration of the acid-etched demineralized dentin. Within the limitations of this investigation, this leads to the belief that the insufficient infiltration may be linked to viscosity and hydrophobicity-hydrophilicity incompatibility of the self-adhesive composite. The micromechanical retention guaranteed by the etching protocol alone, upon collagen exposure, did not allow a correct hybrid layer formation, as hinted by Delgado et al. (2019) [34]. However, there are other material and substrate characteristics that could impact the infiltration into the acid-etched demineralized dentin, and this warrants further research. These include the chemical composition of the composite - specifically the type of monomers and their filler load and the moisture content of acid-etched dentin [47,48]." We also added two new references.
  7. We agree with the reviewer and have now revised the conclusion to be more conservative. In fact, we have listed some problems (more than one) that might explain the results, as suggestions based on the results and their limitations. We believe it is now informative but in the realm of a probability/suggestion, rather than a confirmation which cannot be proven solely by the results of this in vitro study.

Reviewer 3 Report

Dear authors, the reviews are adequate, so I'm considering this study for publication. Thank you for your kind and objective answers.

Author Response

We thank the reviewer for their comment and wish them the best.